METHODS AND PROTOCOLS

# A High-Throughput Yellow Fever Neutralization Assay

Madina Rasulova,[a,b] Thomas Vercruysse,[a,b] Jasmine Paulissen,[a,b] Catherina Coun,[a,b] Vanessa Suin,[c] Leo Heyndrickx,[d] Ji Ma,[a,e] Katrien Geerts,[a,e] Jolien Timmermans,[a,b] Niraj Mishra,[a,e*] Li-Hsin Li,[a,e] Dieudonné Buh Kum,[a,e§] Lotte Coelmont,[a,e] Steven Van Gucht,[c] Hadi Karimzadeh,[f,h] Julia Thorn-Seshold,[f,h] Simon Rothenfußer,[f,h] Kevin K. Ariën,[d,g] Johan Neyts,[a,e] Kai Dallmeier,[a,e] Hendrik Jan Thibaut[a,b]

aKU Leuven Department of Microbiology, Immunology and Transplantation, Rega Institute, Virology and Chemotherapy, Molecular Vaccinology & Vaccine Discovery, Leuven, Belgium

bKU Leuven Department of Microbiology, Immunology and Transplantation, Rega Institute, Translational Platform Virology and Chemotherapy (TPVC), Leuven, Belgium

cSciensano, Viral Diseases Service, Scientific Directorate of Infectious Diseases in Humans, Brussels, Belgium

dVirology Unit, Department of Biomedical Sciences, Institute of Tropical Medicine Antwerp, Antwerp, Belgium

eGlobal Virus Network (GVN), Baltimore, Maryland, USA

fDivision of Clinical Pharmacology, University Hospital, LMU Munich, Munich, Germany

gDepartment of Biomedical Sciences, University of Antwerp, Antwerp, Belgium

hUnit Clinical Pharmacology (EKliP), Helmholtz Center for Environmental Health, Munich, Germany

**ABSTRACT** Quick and accurate detection of neutralizing antibodies (nAbs) against yellow fever is essential in serodiagnosis during outbreaks for surveillance and to evaluate vaccine efficacy in population-wide studies. All of this requires serological assays that can process a large number of samples in a highly standardized format. Albeit being laborious, time-consuming, and limited in throughput, the classical plaque reduction neutralization test (PRNT) is still considered the gold standard for the detection and quantification of nAbs due to its sensitivity and specificity. Here, we report the development of an alternative fluorescence-based serological assay (SNT$^{FLUO}$) with an equally high sensitivity and specificity that is fit for high-throughput testing with the potential for automation. Finally, our novel SNT$^{FLUO}$ was cross-validated in several reference laboratories and against international WHO standards, showing its potential to be implemented in clinical use. SNT$^{FLUO}$ assays with similar performance are available for the Japanese encephalitis, Zika, and dengue viruses amenable to differential diagnostics.

**IMPORTANCE** Fast and accurate detection of neutralizing antibodies (nAbs) against yellow fever virus (YFV) is key in yellow fever serodiagnosis, outbreak surveillance, and monitoring of vaccine efficacy. Although classical PRNT remains the gold standard for measuring YFV nAbs, this methodology suffers from inherent limitations such as low throughput and overall high labor intensity. We present a novel fluorescence-based serum neutralization test (SNT$^{FLUO}$) with equally high sensitivity and specificity that is fit for processing a large number of samples in a highly standardized manner and has the potential to be implemented for clinical use. In addition, we present SNT$^{FLUO}$ assays with similar performance for Japanese encephalitis, Zika, and dengue viruses, opening new avenues for differential diagnostics.

**KEYWORDS** yellow fever virus, serodiagnosis, reporter virus, PRNT, neutralization assay, high-throughput, dengue virus, Zika virus, Japanese encephalitis virus

Yellow fever virus (YFV) is a mosquito-borne, positive-strand RNA virus that belongs to the genus *Flavivirus* within the family of the *Flaviviridae* and is the causative agent of yellow fever (YF). Other clinically important flaviviruses include dengue (DENV), Zika (ZIKV), West Nile (WNV), and Japanese encephalitis (JEV) viruses (1–3). Despite the presence of a safe and very effective vaccine that confers sustained

Address correspondence to Kai Dallmeier, kai.dallmeier@kuleuven.be, or Hendrik Jan Thibaut, hendrikjan.thibaut@kuleuven.be.

*Present address: Niraj Mishra, Gene Therapy Division, Intas Pharmaceuticals Ltd., Ahmedabad, Gujarat, India.

§Present address: Dieudonné Buh Kum, Aligos Belgium, Leuven, Belgium.

The authors declare no conflict of interest.

immunity and long-lasting protection (up to 35 years) in most vaccinees after a single-dose administration, YF still represents a major public health problem throughout the tropical areas of Africa and Central and Southern America (1, 4–6). In addition, insufficient vaccine coverage and international travel rise fear of YFV spreading to the Asian-Pacific regions where the competent mosquito vector is abundantly present and the human population is largely immunologically naive to YFV (7–10).

In the early and acute stages of disease, YF diagnosis is based on assessing patient's clinical features in combination with conventional (endpoint) or real-time reverse transcription-PCR (RT-PCR) (1, 11), viral isolation, or, in fatal conditions, immunohistochemical analysis to detect YF antigens in liver and other postmortem tissues (11, 12). In the later stages of infection, several serological methods are used to diagnose YF. Due to its simplicity to detect YFV-specific immunoglobulin M (IgM) and/or immunoglobulin G (IgG) antibodies, enzyme-linked immunosorbent assays (ELISA) have become the primary diagnostic tool worldwide (11, 13–16). However, cross-reactivity with other flaviviruses or nonspecific reactivity often represents a major disadvantage in evaluating the infection in areas where other flaviviruses cocirculate (especially dengue and Zika viruses) (11). Therefore, the more specific and sensitive plaque reduction neutralization test (PRNT), developed more than 5 decades ago (17), is currently still recommended as the "gold-standard" assay worldwide. However, this assay suffers from several major disadvantages, including its duration, labor intensity, unsuitability for high-throughput settings, and the requirement of highly qualified and experienced staff to manually count plaque numbers (18).

Because of these technical drawbacks, there is an urgent need for a rapid, highly specific, and robust surveillance and diagnostic tool. Such a method would help to recognize YFV outbreaks in an earlier stage, ease testing people with suspicion of YFV infection living in or returning from regions of endemicity, and, hence, prevent viral spreading. In addition, in 2017, the WHO has launched a global strategy aiming to eliminate YF epidemics (EYE) by 2026 through vaccination of 1.4 billion people residing in YF areas of endemicity and to contain outbreaks rapidly by the use of a fractional dose of the YF17D vaccine (one-fifth of the normal dose) (19). As it is still unclear how this fractional dosing will affect the lifelong protection provided by YF17D (20, 21), large population-wide studies are required to monitor vaccine efficacy. Finally, during vaccine efficacy studies, it is also key to understanding the impact of a new vaccine candidate on the effectiveness of existing vaccines, including YF17D (22–32). As such extensive studies require the processing of a large number of serum samples, it is of essence to have a serological assay at hand that is robust and specific with improved turnaround and throughput properties.

Here, we describe an easy-to-use diagnostic method to quantify neutralizing antibodies using a fluorescently tagged YF17D as an alternative to classical PRNT. Our reporter-based neutralization assay is amenable to high-throughput screening (HTS) with the possibility of automation, allowing detection of neutralizing antibodies in a large number of serum samples. This qualifies it as a powerful diagnostic tool for rapid identification of ongoing YFV outbreaks and assessment of vaccination efficiency in clinical trials, especially if only small serum volumes are available. Furthermore, quantification of neutralizing activity is fully automated, generating less subjective data than any manual counting method.

## RESULTS

**Development of a high-throughput fluorescence-based seroneutralization test.** To increase the speed of YF serodiagnosis, it is key to implement a rapid and highly specific neutralization assay that is amenable to high-throughput screening and is compatible with automated readout and data analysis, ideally requiring only a limited volume of serum sample. Therefore, we developed an alternative fluorescence-based seroneutralization test (SNT[FLUO]) to rapidly and reliably quantify YFV-neutralizing antibodies in sera (Fig. 1). To this end, we generated virus from a plasmid engineered to express YF17D together with

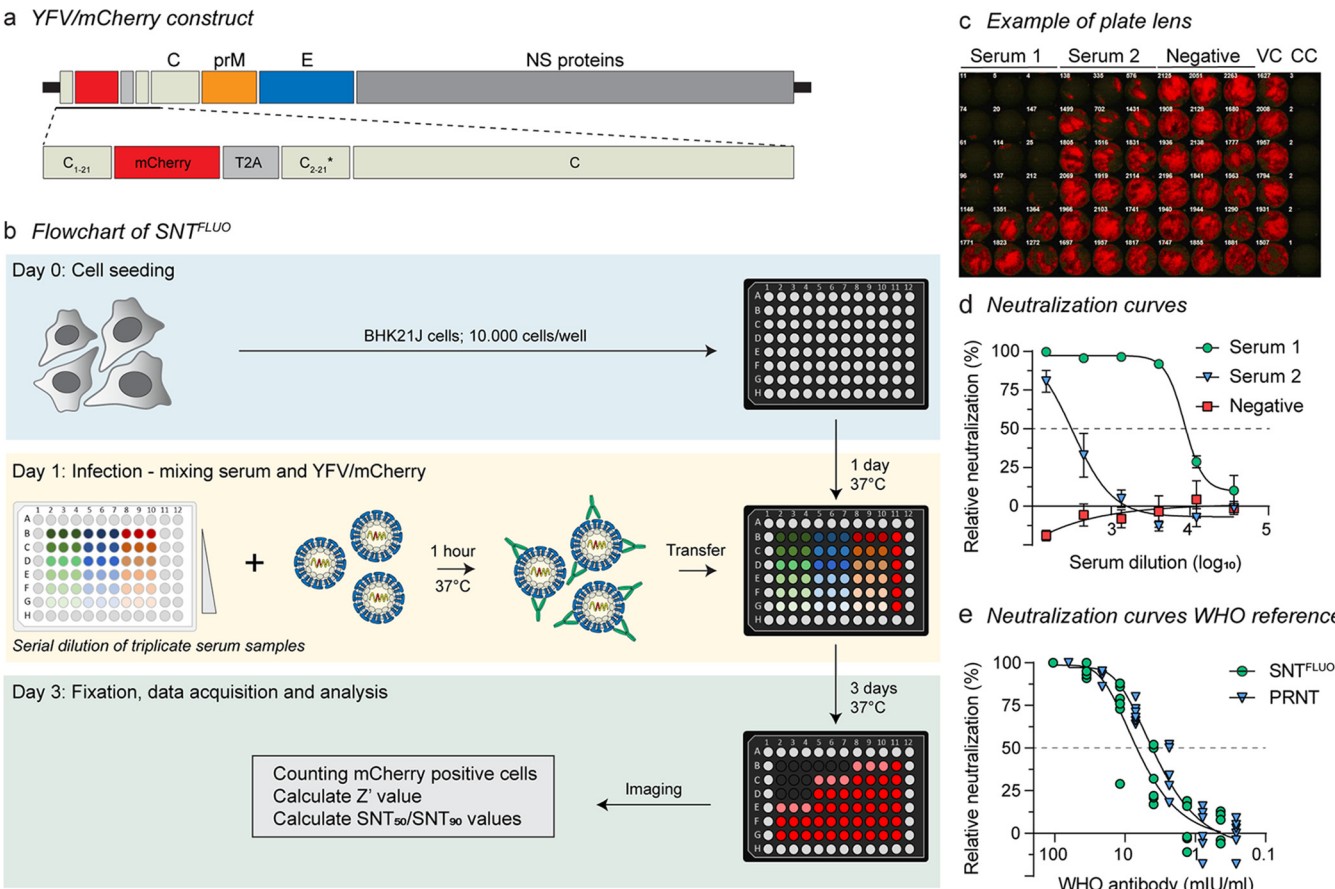

**FIG 1** A high-throughput fluorescence-based seroneutralization assay for YFV diagnostics. (a) Schematic representation of YFV-mCherry. The reporter mCherry gene was inserted immediately downstream of codon 21 of the YF17D C gene and flanked by a *Thosea asigna* virus self-cleaving 2A peptide at the 3′ end, followed by a repeat of C gene codons with an alternative sequence (C2-C21*). (b) Assay flowchart of SNT$^{FLUO}$. A detailed step-by-step bench protocol is provided as an extended data file. YFV-mCherry was coincubated for 1 h with serially diluted sera in triplicate prior to infecting preseeded BHK-21J cells in a 96-well plate. Three days postinfection, cells were fixed, and the fluorescent spots of infected cells were quantified and analyzed using a CTL ImmunoSpot reader and Genedata Screener software tool, respectively. (c and d) Representative image and neutralization curves of a counted plate containing 2 positive and 1 negative serum samples. (e) Neutralization curves obtained by SNT$^{FLUO}$ and PRNT on the WHO reference sample. Data are means ± standard deviations of three (d) or six (e) replicates.

mCherry (YFV-mCherry) as a translational fusion to the C protein (Fig. 1a) (33, 34). The mCherry transgene remained stable up to at least six passages as demonstrated by RT-PCR fingerprinting performed on serially passaged YFV-mCherry in BHK-21J cells (see Fig. S1a and b in the supplemental material).

As a first step toward assay setup, serially diluted YFV-mCherry was used to infect BHK-21J cells to identify the optimal balance between virus input (i.e., number of fluorescent cells, or spots), assay robustness (i.e., Z′ prime value) and assay endpoint (Fig. S1c). On day 3, the highest Z′ value (i.e., >0.5) was obtained at the lowest virus dilution, immediately below a saturation point (i.e., ~2,000 spots/well of a microtiter plate) and corresponding to a multiplicity of infection (MOI) of 0.02 as determined by classical plaque assay. Latter parameters were chosen for further assay development and validation.

A flowchart for our SNT$^{FLUO}$ in a 96-well format is depicted in Fig. 1b. A detailed step-by-step bench protocol is provided as an extended data file. Briefly, sera were serially 1:3 diluted in triplicate in round-bottom 96-well plates and coincubated with YFV-mCherry for 1 h at 37°C. Subsequently, the virus-serum mixtures were added to preseeded BHK-21J cells and further incubated for another 3 days. After fixation, the number of spots was quantified using a CTL ImmunoSpot S6 Ultimate reader (Fig. 1c), and data were analyzed using Genedata Screener software package. In this way, assay statistics and dose-response curves are automatically calculated to determine the serum dilution fold that reduces 50%

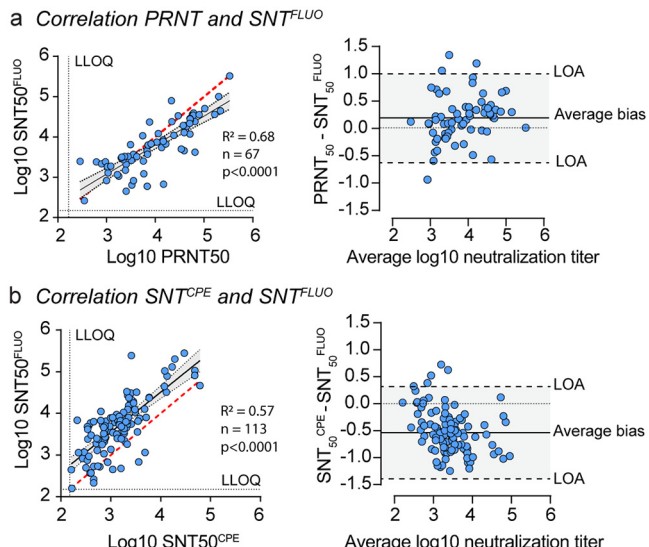

**FIG 2** Benchmarking against other seroneutralization assays. (a and b) Correlation analysis of PRNT and SNT$^{FLUO}$ (a) and SNT$^{CPE}$ and SNT$^{FLUO}$ (b). Left panels show linear regression analysis to calculate correlation coefficients. The Pearson correlation efficient, $R^2$, represents number of tested sera ($n$), and $P$ values are indicated. Perfect correlation is indicated by the red dashed line, whereas correlation between SNT$_{50}^{FLUO}$ and PRNT$_{50}$ or SNT$_{50}^{FLUO}$ and SNT$_{50}^{CPE}$ is indicated by a black solid line. Ninety-five percent confidence intervals are indicated by gray shaded areas. LLOQ, lower limits of quantification. Right panels show Bland-Altman analysis to estimate the degree of agreement between assays. Differences between PRNT$_{50}$ and SNT$_{50}^{FLUO}$ and SNT$_{50}^{CPE}$ and SNT$_{50}^{FLUO}$ values are compared with their average log$_{10}$ neutralization titer. The lines of no bias (dotted line), average bias (solid line), and 95% confidence intervals (i.e., lower and upper limits of agreements [LOA; dashed lines]) are shown. Data are means from three replicates.

of the number of spots (SNT$_{50}$) (Fig. 1d). To accurately quantify the number of infected cells, we analyzed our SNT$^{FLUO}$ assay both by ImmunoSpot reader and high content imaging. On the same surface, we determined that a single spot measured by ImmunoSpot reader corresponds to, on average, 2.5 infected cells, as measured by high-content imaging analysis.

For further initial validation and benchmarking of our reporter assay, we performed both SNT$^{FLUO}$ and conventional PRNT on the WHO international reference sample standardized monkey serum preparation containing YFV nAb (35) for a direct head-to-head comparison of the respective neutralization results (Fig. 1e). Fifty percent effective dose (EC$_{50}$) values obtained by either assay were very much comparable and in the same range: they were 6.4 mIU/mL (95% confidence interval, 5.2 to 7.8) and 4.0 mIU/mL (95% confidence interval, 3.4 to 4.7) for SNT$^{FLUO}$ and PRNT, respectively, as the first evidence of similar performance of SNT$^{FLUO}$ and PRNT. Finally, to evaluate assay reproducibility and construct stability, we performed seven parallel SNT$^{FLUO}$ assays using YFV-mCherry from different passages (P0, i.e., harvest after transfection, and P1 to P6), showing similar neutralization results (Fig. S1d) (mean titer, 4.7 ± 1.7 mIU/mL), with SNT$_{50}$ values remaining comparable to those obtained by PRNT. In conclusion, our results show that the YFV-mCherry reporter assay could serve as an equally sensitive, easy-to-use alternative for the gold-standard PRNT.

**Assay validation by benchmarking against other neutralization tests.** To further validate our SNT$^{FLUO}$, we quantified neutralizing antibody levels in a large set of historical serum samples from previous vaccination studies with YF17D in mice, hamsters, and nonhuman primates (NHP). For this purpose, we used several methods, including the here-described SNT$^{FLUO}$, conventional PRNT, and another 96 well-based neutralization assay that uses cytopathic effect (CPE) as readout (SNT$^{CPE}$) (33, 36). PRNT and SNT$^{FLUO}$ were performed on 67 serum samples (Fig. 2a) and SNT$^{CPE}$ and SNT$^{FLUO}$ on 113 serum samples (Fig. 2b), including only samples with nAb titers above the lower limit of quantification (LLOQ) in both assays for calculation of correlation coefficients in

**FIG 3** $SNT^{FLUO}$ cross-validation by reference laboratories. YF17D-vaccinated ($n = 6$) and unvaccinated ($n = 4$) monkey sera were serially 1:10 diluted ($1:10^1$ to $1:10^4$) and assessed for the presence of neutralizing antibodies by the three different laboratories using their respective assays. The $EC_{50}$ values for each serum sample between the upper and lower quantification limits were used for correlation analysis (see Table S1 in the supplemental material). Data were analyzed by linear regression to calculate correlation coefficients. The Pearson correlation efficient, $R^2$, indicates number of tested sera ($n$), and $P$ values are indicated. Perfect correlation is indicated by a red dashed line, whereas correlation between the different assays is indicated by a black solid line. Ninety-five percent confidence intervals are indicated by gray shaded areas. LLOQ, lower limits of quantification. Data are the means of six (ITM and our laboratory) or two (Sciensano) replicates.

regression analysis. The strongest correlation was observed between $PRNT_{50}$ and $SNT_{50}^{FLUO}$, with an $R^2$ value of 0.68 ($n = 67$) (Fig. 2a, left). A weaker correlation was observed between $SNT_{50}^{FLUO}$ and $SNT_{50}^{CPE}$, with an $R^2$ value of 0.57 ($n = 113$) (Fig. 2b, left). Bland-Altman analysis was used to estimate the degree of agreement between assays and reveal any possible bias between their mean differences (37). In line with our regression analysis, the smallest bias and 95% confidence intervals (i.e., limits of agreement [LOA]) were found between $PRNT_{50}$ and $SNT_{50}^{FLUO}$ values, with an average bias of $0.19 \pm 0.42$ (LOA from $-0.63$ to 1.0) (Fig. 2a, right), whereas a markedly increased bias of $-0.54 \pm 0.43$ (LOA from $-1.4$ to 0.31) was observed in the comparison of the $SNT_{50}^{CPE}$ and $SNT_{50}^{FLUO}$ values (Fig. 2b, right). These results further confirm that $SNT^{FLUO}$ yields neutralization results that are comparable to those obtained by PRNT.

**Assay cross-validation by reference laboratories.** A set of serum samples was prepared for cross-validation by the National Reference Center for Arboviruses at the Institute of Tropical Medicine in Antwerp (PRNT; ISO 15189) (38) and Sciensano, Viral Diseases Service in Brussels (rapid fluorescent focus inhibition test [RFFIT]; ISO 17025) (39), two accredited reference laboratories in Belgium. To this end, serum samples from nonvaccinated ($n = 4$) and YF17D-vaccinated ($n = 6$) NHPs were serially 1:10 diluted ($1:10^1$ to $1:10^4$) and assessed blindly for the presence of nAbs in the three different laboratories using their respective assays (Table S1 and Fig. 3). Six technical repeats of 34 serum samples in total were used to obtain dose-response curves from which 50% neutralizing activities ($EC_{50}$) were calculated. For correlation analysis between the three different assays, all nAb titers within the respective lower and upper quantification limits of each sample were included (Table S1). The strongest correlation was observed between $SNT_{50}^{FLUO}$ and $PRNT_{50}$ values (Fig. 3) with $R^2$ of 0.85 ($n = 11$), whereas $R^2$ of 0.78 was observed between $SNT_{50}^{FLUO}$ and $RFFIT_{50}$ ($n = 8$). These results indicate that the $SNT^{FLUO}$ has the potential to be implemented in reference laboratories, including for high-throughput serodiagnostics, as a highly sensitive alternative for conventional time-consuming assays.

**Assay specificity.** The assay specificity of $SNT^{FLUO}$ was assessed using four groups of potentially cross-reactive sera from previous studies in which mice were vaccinated for either YF17D ($n = 8$), Japanese encephalitis virus (JEV; $n = 8$), Zika virus (ZIKV; $n = 9$), or dengue type 2 virus (DENV2; $n = 5$) (Fig. 4a). All 30 serum samples were assessed for specificity and cross-reactivity using similar fluorescence-based $SNT^{FLUO}$ assays by using four distinct reporter viruses, YFV-mCherry, DENV2-mCherry, ZIKV-mCherry, and

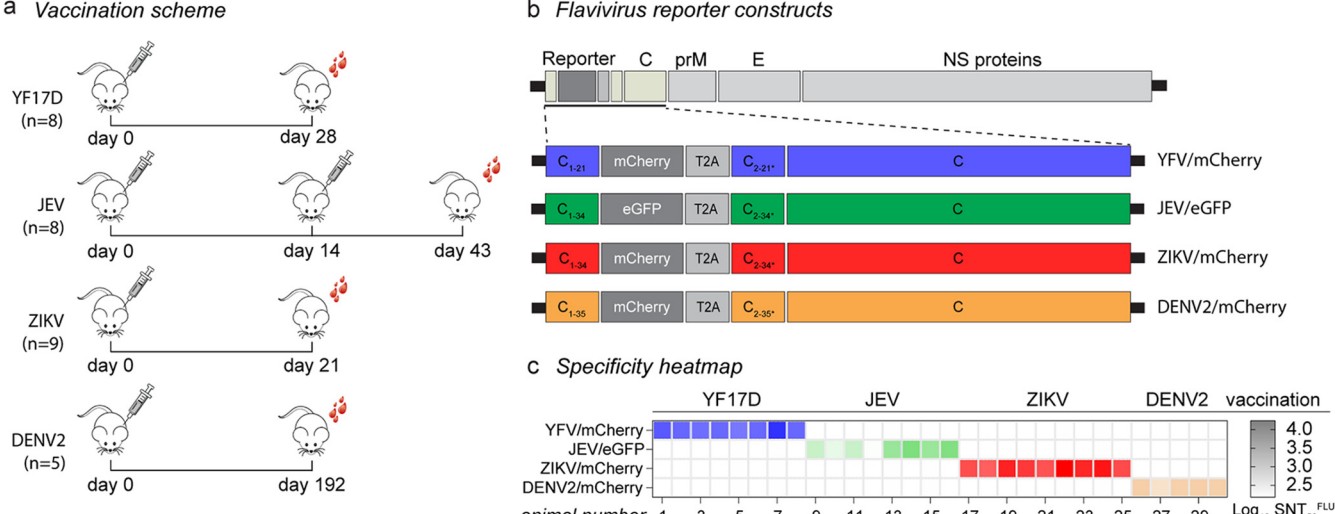

**FIG 4** Specificity of SNT^FLUO against other flaviviruses. (a) Schematic representation of mice vaccination scheme for YF17D (*n* = 8), JEV (*n* = 8), ZIKV (*n* = 9), and DENV2 (*n* = 5), and day of serum collection (total of 30 serum samples). (b) Schematic representation of YFV-mCherry, JEV-eGFP, ZIKV-mCherry, and DENV2-mCherry. Reporter viruses were generated in a similar way as for YFV-mCherry, with the exception of extended and alternative codons of the C gene, respectively, flanking both ends of the fluorescent tag sequence (34 for JEV and ZIKV and 35 for DENV2). (c) Specificity heatmap of neutralizing titers (SNT$_{50}$^FLUO) of the 30 serum samples tested, using the 4 reporter flaviviruses. Data are the means of three replicates. Heatmap gradient indicates log$_{10}$ SNT$_{50}$^FLUO values.

JEV-eGFP (Fig. 4b). DENV2-mCherry (40) has been described earlier; ZIKV-mCherry and JEV-eGFP were generated accordingly for this study. All test sera specifically neutralized only their corresponding cognate reporter virus without detectable cross-neutralizing activity toward the other flavivirus reporters (Fig. 4c and Fig. S2). These data indicate that YFV-mCherry SNT^FLUO is highly specific without notable flavivirus cross-reactivity. Furthermore, a panel of similar fluorescence-based neutralization tests may be added to the current diagnostic repertoire to aid rapid serological identification and differentiation of flavivirus infections.

**Implementation of SNT^FLUO for high-throughput screening.** To explore whether our novel SNT^FLUO approach is fit for high-throughput serodiagnosis, we tested approximately 2,000 serum samples from different organisms (867 mice, 132 hamsters, 96 pigs, 174 monkeys, and 727 humans), either unvaccinated or vaccinated with YF17D. In total, more than 600 plates were analyzed for robustness by calculating Z′ values, percent coefficients of variation (CV%), and signal-to-background ratios (S/B) (Fig. 5a). The Z′ value achieved for the majority (>90%) of the plates was >0.5, with a variability of <20% and S/B between 200 and 3,500. About 1% of plates with a Z′ of <0.1 needed to be rejected, whereas plates with a Z′ value between 0.1 and 0.5 (7%) were subjected to visual inspection (by fluorescence microscopy) to identify outliers, or possibly, be rejected (Fig. 5a, left, orange and red circles). Notably, poor assay robustness corresponded with a decrease in virus titer (reduced spot counts in untreated virus control wells), as expected for YF17D stocks during long-term storage at −80°C (Fig. S1c), requiring back titration and adjustment of the infectious input material as performed during initial assay setup. Finally, the suitability of SNT^FLUO to monitor YF17D vaccination efficiency in larger population-wide studies was assessed. Therefore, we quantified the levels of nAb in matched serum samples (total of 727 sera) from a large cohort of subjects prior to (*n* = 249) and after YF17D vaccination (*n* = 478) at 2 different time points (Fig. 5b). Additionally, these 249 prior vaccination samples were used to accurately determine the limit of detection of our assay (i.e., 1:10). Altogether, the SNT^FLUO assay was capable of reliably and rapidly diagnosing the absence, or likewise, seroconversion to YFV-specific nAbs, further demonstrating its potential to be used in a clinical setting where a fast sample turnaround is highly desirable.

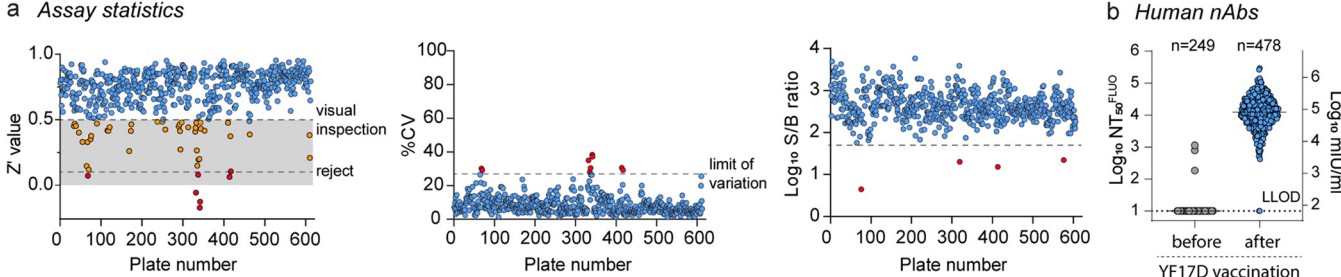

**FIG 5** High-throughput performance of SNT$^{FLUO}$. (a) We analyzed 600 assay plates containing approximately 2,000 serum samples for assay robustness by calculating Z′ values (left), percent coefficients of variation (CV%; middle), and signal-to-background ratios (S/B; right). Gray area in the left panel indicates Z′ score between 0 and 0.5. Plates with a Z′ score between 0.5 and 0.1 were subjected to visual inspection by fluorescence microscopy (orange circles). Plates with a Z′ score of <0.1 were rejected (red circles). (b) YF17D vaccination efficiency in a larger population-wide study prior to ($n = 249$) and postvaccination ($n = 478$). Blood was collected on day 14 and day 28 postvaccination and assessed for its neutralizing activity. Data are presented as log$_{10}$ SNT$_{50}$$^{FLUO}$ (left $y$ axis) or transformed to log$_{10}$ mIU/mL (right $y$ axis). Data are the means of three replicates. LLOD, lower limits of detection.

## DISCUSSION

Fast and accurate quantification of YFV-specific nAbs plays a key role in YFV serodiagnosis, surveillance of large cohorts, and population-wide monitoring of vaccine immunogenicity. Currently, the highly sensitive and specific PRNT assay is considered the gold standard for evaluating the presence of YFV-specific nAbs, but it suffers from several drawbacks. In this study, we developed a fluorescence-based yellow fever virus neutralization (SNT$^{FLUO}$) assay as a high-throughput, rapid, and easy-to-use alternative to the traditional PRNT.

We characterized the sensitivity and specificity of the YFV-mCherry reporter SNT$^{FLUO}$ by using a WHO reference standard and more than 2,000 historical serum samples originating from several species backgrounds, including mice, hamsters, pigs, NHPs, and humans. A head-to-head comparison of SNT$_{50}$ and PRNT$_{50}$ values obtained by SNT$^{FLUO}$ and PRNT, respectively, further demonstrated similar sensitivity and good sample-to-sample correlation between both assays. Cross-validation of a set of NHP serum samples by two accredited reference laboratories in Antwerp (ITM) and Brussels (Sciensano) further underlined the possibility of adding the SNT$^{FLUO}$ to the current YF serodiagnosis assay repertoire.

Due to their complexity, PRNT assays are currently confined solely to accredited reference laboratories (14). To overcome this bottleneck, there is a big need for an assay amenable to automation, including a fully automated data analysis pipeline. To this end, we used a fluorescent reporter yellow fever virus expressing an mCherry reporter, which was chosen because of its small monomeric size, excellent brightness, and superior photostability (41). These properties render the SNT$^{FLUO}$ assay highly suitable for fluorescence-based quantification by automated imagers. In addition, by using predetermined scanning and counting settings, user-dependent biases are kept to a minimum. In parallel, assay performance statistics are automatically calculated, allowing the accurate monitoring of assay quality.

Compared with the PRNT assay, our SNT$^{FLUO}$ has shortened the assay duration to 4 days, decreased the volume of serum that is required (from 50 $\mu$L to 26 $\mu$L), facilitated assay manipulation (e.g., by omitting overlays), and increased the testing capacity to high throughput. Although this assay has been developed for 96-well plate format, upscaling to a 384-well plate format should be possible, further increasing its throughput.

Another advantage of our SNT$^{FLUO}$ assay is the ease of sharing reagents with other laboratories. Indeed, virus stocks can be produced from pShuttle-YFV-mCherry plasmid transfection, followed by passaging to produce larger YFV-mCherry stocks that stably express the mCherry transgene for at least six passages while retaining sensitivity of the fluorescence signal to neutralization. Since plasmids are highly stable and can be easily shipped to other laboratories at ambient temperature, the stability of the construct in use and hence the assay can be ensured.

In addition to its high sensitivity, we also showed that the YFV-mCherry SNT$^{FLUO}$

assay exhibits exceptional specificity with no detectable cross-neutralization by potentially cross-reactive sera from animals vaccinated for either ZIKV, DENV2, or JEV. However, the level of possible cross-reactivity and cross-neutralization still needs to be explored in a clinical setting where multiple heterotypic infections can occur. In addition, we report on several other flavivirus reporter-based SNT$^{FLUO}$ assays, including for ZIKV, DENV2, and JEV. Similar to YFV-mCherry, no cross-reactivity was observed between the different sera. Although these assays still need to be further characterized and validated, our data already suggest that they could be implemented in a similar way as the YFV-mCherry SNT$^{FLUO}$ assay. The availability of such assays with no cross-reactivity could be of great benefit for differential serology to accurately and rapidly identify a specific flavivirus infection, which is especially challenging in regions where several flaviviruses cocirculate. Finally, similar reporter assays could be optimized toward high-throughput antiviral screening campaigns, either as single (40, 42) or, by choice of compatible fluoroprotein reporters, multiplexed antiviral screens, possibly allowing identification of pan-flavivirus antivirals.

In conclusion, our SNT$^{FLUO}$ assay offers a powerful tool to rapidly and specifically identify YFV-specific nAb. Furthermore, its sensitivity, robustness, and suitability for high-throughput screening make it a valuable alternative to conventional PRNT.

## MATERIALS AND METHODS

**Cells.** Baby hamster kidney fibroblasts (BHK-21J) and African green monkey kidney (Vero E6) cells were maintained in minimum essential medium (MEM; Gibco) supplemented with 10% fetal bovine serum (HyClone), 2 mM L-glutamine (Gibco), 1% sodium bicarbonate (Gibco), 1× MEM-nonessential amino acids solution (Gibco), 10 mM HEPES (Gibco), and 100 U/mL penicillin-streptomycin (Gibco) (here referred as seeding medium) and incubated at 37°C. All assays were performed in the same medium but containing 2% fetal bovine serum (assay medium).

**Plasmid construction.** Plasmid construction of the yellow fever 17D vaccine strain (pShuttle-YFV-mCherry) and dengue virus type 2 NGC strain (pShuttle-DENV2-mCherry) reporter viruses stably expressing the red fluorescent protein mCherry have been reported previously (33, 40). In a similar manner, Zika (pShuttle-ZIKV-mCherry) and Japanese encephalitis (pShuttle-JEV-enhanced green fluorescent protein [eGFP]) reporter viruses were generated for the current study (schematic representation in Fig. 1 and 4). In brief, cDNA of ZIKV strain BeH819015 (43) was first inserted in pShuttle. This construct, together with the previously described pShuttle plasmid containing JEV strain SA14-14-2 (33), was next used to generate ZIKV and JEV reporter viruses expressing mCherry or eGFP, respectively, as a translational fusion to the N terminus of the C protein. Using standard molecular biology techniques and homologous recombination in yeast (strain YPH500), the synthetic DNA fragments encoding codons 2 to 236 of mCherry (GenBank accession no. AY678264) or codons 1 to 238 of eGFP (GenBank accession no. HI137399) were inserted immediately downstream of codon 34 of the C gene of the corresponding virus. The reporter gene is flanked by a BamHI restriction site at its 5′ terminus and by the ribosome-skipping 2A sequence of *Thosea asigna* virus (44) at the 3′ end. C gene codons 2 to 34* were repeated with an alternative codon usage to avoid recombination during virus replication. The plasmids were recovered from yeast and transformed into Epi300 (Epicenter)-competent *E. coli* cells, and colonies were selected as described earlier (40). The entire genome of ZIKV-mCherry and JEV-eGFP was verified by Sanger sequencing.

**Viruses and virus titrations.** Infectious viruses were rescued from plasmid constructs by transfection into BHK-21J (pShuttle-YF17D, pShuttle-YFV-mCherry, pShuttle-JEV-eGFP, and pShuttle-DENV2-mCherry) and Vero E6 (pShuttle-ZIKV-mCherry) cells using TransIT-LT1 transfection reagent (Mirus Bio) following the manufacturer's instructions (33, 40). Upon onset of cytopathic effect (CPE), the recombinant viruses were subsequently passaged on their corresponding cell lines to generate virus stocks. DENV2-mCherry was generated in C6/36 as described earlier (40). The harvested supernatants were centrifuged at 2,100 × *g* for 8 min, aliquoted, and stored at −80°C. To determine the stability of mCherry insert in the YF17D backbone, the reporter virus was passaged up to passage 6 on BHK-21J. Virus stocks of passages 4 and 5 were used for all the experiments performed in this study.

YFV-mCherry, DENV2-mCherry, and JEV-eGFP were titrated on BHK-21J cells; ZIKV-mCherry were titrated on Vero cells. In parallel, we determined the infectious virus titer for YFV-mCherry, DENV2-mCherry, and JEV-GFP by classical plaque assay. Due to an increased attenuation by the presence of the mCherry transgene, ZIKV-mCherry does not induce measurable CPE either by plaque or 50% tissue culture infective dose (TCID$_{50}$) assay. Briefly, cells were seeded in 96-well black plates with transparent bottoms (Greiner Bio-One) in assay medium and incubated overnight at 37°C. The next day, cells were inoculated with serial 3-fold dilutions of the reporter viruses and further incubated for 3 days. Here, cells were fixed with 4% formaldehyde for 30 min at room temperature (RT) and washed once with Dulbecco's phosphate-buffered saline (DPBS; Gibco). The plates were air-dried in the dark for 30 min prior to analysis. Scanning and counting of fluorescent spots were performed in an ImmunoSpot reader using FluoroSpot and BioSpot modes (S6 Ultimate software; Cellular Technology Limited). Z′ values of each viral dilution were determined using the following formula:

$$Z' = 1 - \frac{3\sigma_{\text{VD}} + 3\sigma_{\text{CC}}}{|\mu_{\text{VD}} - \mu_{\text{VD}}|}$$

Here, $\sigma$ and $\mu$ stand for standard deviation and mean of spots, respectively, detected in either a certain virus dilution (VD) or cell control (CC).

**Viral RNA isolation and reverse transcription-PCR.** Total viral RNA was extracted from 0.1 mL virus stocks using Aurum Total RNA minikit (Bio-Rad) and eluted in 50 $\mu$L elution buffer preheated to 70°C. The generation of cDNA and amplification of mCherry region were carried out using qScript XLT one-step RT-PCR kit (Quanta Bioscience). RT-PCR conditions were as follows: RT step at 55°C for 20 min, initial denaturation at 94°C for 3 min, 40 cycles of amplification (denaturation at 94°C for 30 s, annealing at 60°C for 45 s, and elongation at 72°C for 50 s), and final extension at 72°C for 10 min. The sequences of the primers were YFV/mCherry-Forward, 5'-GCAAATCGAGTTGCTAGGC-3', and YFV/mCherry-Reverse, 5'-CTTGAACACCTCTTGAAGG-3'. Each primer was used at a final concentration of 600 nM.

**Serum neutralization test.** A detailed protocol is available as an extended data file. Briefly, sera were serially 1:3 diluted in triplicate in round-bottom 96-well plates and coincubated with YFV-mCherry for 1 h at 37°C. The inoculum titer was predetermined by a titration experiment to result in 1,500 to 2,000 spot-forming units and corresponds to MOIs of 0.02, 0.04, and 4.5 for YFV-mCherry, JEV-GFP, and DENV2-mCherry, respectively. As described above, ZIKV-mCherry infection does not induce measurable CPE. After 1 h, the "serum-virus" complexes were added in triplicate to preseeded cells and incubated at 37°C for 1 h. Additionally, each 96-well plate contained six wells with YFV-mCherry-infected cells, and six wells with uninfected cells serving as positive and negative controls, respectively. Fixation and imaging were performed similarly as described above in "Viruses and virus titrations." The titers of nAbs protecting 50% ($SNT_{50}$) and 90% ($SNT_{90}$) of the cells from YFV-mCherry infection were determined by fitting the serum neutralization-dilution curve that is normalized to the virus infection (100%) and cell control (0%) using Genedata Screener version 17.0.4. The limit of quantification was defined by the lowest dilution used in the assay; the limit of detection was determined by calculating the average neutralization titer multiplied by three times the standard deviation of 253 negative serum samples.

**Plaque reduction neutralization test.** The PRNT assay was performed as described previously [33]. Briefly, BHK-21J cells were seeded in 12-well plates (Falcon) in seeding medium and cultured overnight at 37°C. The next day, triplicate serial dilutions of serum were coincubated with YF17D for 1 h at 37°C prior to addition to BHK-21J cells. After 1 h of incubation, cells were washed twice with assay medium and overlaid with 2× Temin's modified eagle medium (Gibco) supplemented with 4% fetal bovine serum (FBS) and 0.75% sodium bicarbonate containing 0.5% low-melting agarose (Invitrogen). The overlay was allowed to solidify at RT; cells were then cultured for 5 days at 37°C, fixed with 8% formaldehyde, and visualized by staining with methylene blue. Plaques were manually counted, and $PRNT_{50}$ values were calculated by either serum dilution curve fitting using Genedata Screener version 17.0.4 or a non-linear fitting algorithm [log(agonist) versus response − variable slope (four parameters)] in GraphPad version 8. The limit of quantification was defined by the lowest dilution used in the assay.

**Plaque reduction neutralization test.** The certified PRNT assay (ISO 15189) was performed in 96-well plates (Falcon) using 2-fold serum dilutions (ranging from 1:10 to 1:320) in six replicates. Briefly, serially diluted sera were coincubated with predefined YF17D titer for 1 h at 37°C. Next, porcine kidney (PS) cells were added in Dulbecco's modified Eagle's medium (DMEM) supplemented with 5% FBS, 1% penicillin-streptomycin, and 1% L-glutamine to the serum-virus mix and incubated for 3 to 4 h. Cells were overlaid with 1.2% Avicel in DMEM supplemented with 5% FBS and incubated for 4 days at 37°C and 7% $CO_2$, fixed with 3.7% formaldehyde, and visualized by staining with naphtol blue/black. Plaques were manually counted, and $PRNT_{50}$ values were calculated using the Reed-Muench method (Reed & Muench, 1938) [45].

**Cell-based CPE assay and CPE-based virus neutralization test.** The $SNT^{CPE}$ assay was performed as described previously [33]. Briefly, BHK-21J cells were plated in 96-well transparent plates (Corning) in seeding medium and incubated overnight at 37°C. The next day, triplicate serial dilutions of serum were coincubated with 100 $TCID_{50}$ of YF17D for 1 h at 37°C and then added to the cells. After 5 days of incubation at 37°C, each well was first visually inspected for the signs of CPE and then stained using 3-(4,5-dimethylthiazol-2-yl)-5-(3-carboxymethoxyphenyl)-2-(4-sulfophenyl)-2H-tetrazolium-phenazine methosulfate (MTS-PMS) (Merck) for one to 2 h at 37°C followed by absorbance reading at the 498-nm wavelength using a microtiter plate reader (Safire, Tecan). $TCID_{50}$/mL was determined by serum dilution curve fitting using Genedata Screener version 17.0.4. The limit of quantification was defined by the lowest dilution used in the assay.

**Rapid fluorescent focus inhibition test.** The rapid fluorescent focus inhibition test (RFFIT) assay (ISO 17025) was performed as described previously (Roelandt et al., 2016) [46]. Briefly, sera, including positive and negative controls, were 3-fold diluted (ranging from 1:9 to 1:243) in DMEM (Gibco) supplemented with 10% heat-inactivated fetal calf serum (FCS; Gibco). YFV strain French neurotropic (National Collection of Pathogenic Viruses) was added at a dose of approximately 1.2-log $TCID_{50}$ to the wells containing the diluted sera. Following this, BHK-21 cells were added to each well and incubated for 24 h at 37°C and 5% $CO_2$. Plates were fixed with 100% methanol at 4°C for 30 min. Infected BHK-21 cells were detected by an indirect immunofluorescence staining, using a primary mouse monoclonal antibody against the YFV envelope protein (Abcam) and a secondary Alexa Fluor 488-conjugated goat anti-mouse IgG antibody (Molecular Probes). The number of foci with infected cells was counted under the fluorescence microscope. The serum neutralization (SN) titer was defined as the dilution of test serum that neutralized 50% of the virus ($DIL_{50}$), calculated according to the Reed-Muench method (Reed and Muench, 1938).

**Ethics statement.** Mouse and hamster sera used were sourced from historical samples from immunization experiments conducted at the KU Leuven Rega Institute in accordance with institutional guidelines

approved by the Ethical Committee of the KU Leuven, Belgium. Monkey sera were sampled from purpose-bred rhesus macaques (*Macaca mulatta*) housed at the Biomedical Primate Research Centre (BPRC, Rijswijk, The Netherlands) and vaccinated with either Stamaril (Sanofi-Pasteur) or mock upon positive advice by the independent ethics committee (DEC-BPRC) under project license DEC753C issued by the Central Committee for Animal Experiments according to Dutch law. Human sera before and after vaccination with the YFV vaccine Stamaril were derived from a YF17D vaccination study, approved by the ethical committee of the Medical Faculty, LMU Munich (IRB number 86-16).

**Serum samples and controls.** Mice, hamsters, pigs, monkeys, and humans vaccinated for YF17D, JEV, ZIKV, or DENV2 were collected for monitoring by other studies independent of this study. Monkey sera used for cross-validation of SNT$^{FLUO}$ at the two Belgium reference centers were collected from rhesus macaques vaccinated with licensed YF17D vaccine Stamaril. Monkey anti-YF serum calibrated by a WHO international reference preparation (143 IU/mL) was used as a positive control (35) both in PRNT and SNT$^{FLUO}$. To inactivate complement, all sera were heat inactivated by incubation at 56°C for 30 min prior to use.

**Statistical analysis.** GraphPad Prism (GraphPad Software, version 8) was used for all statistical evaluations. The number of animals or humans and number of replicate experiments that were performed are indicated in the figure legends. Correlation studies were performed using linear regression analysis with Pearson's correlation coefficient and Bland-Altman analysis. Values were considered statistically significantly different at $P$ values of $\leq 0.05$.

**Data availability.** The data sets generated and/or analyzed during the current study are available from the corresponding authors upon reasonable request.

## SUPPLEMENTAL MATERIAL

Supplemental material is available online only.

**SUPPLEMENTAL FILE 1**, PDF file, 1.1 MB.

## ACKNOWLEDGMENTS

We thank Peter Bredenbeek (LUMC, The Netherlands) for providing BHK-21J and Vero E6 cells and Andres Merits (University of Tartu, Estonia) for sharing cDNA of the ZIKV strain. We also thank Javarappa Mahadesh Prasad Arkalagud for providing serum samples from ZIKV-vaccinated mice. Ernst Verschoor and Babs Verstrepen (Biomedical Primate Research Centre, BPRC Rijswijk, The Netherlands) are acknowledged for providing nonhuman primate sera.

This project has received funding from the Research Foundation Flanders (FWO) under the Excellence of Science (EOS) program (no. 30981113; VirEOS), the European Union's Horizon 2020 research and innovation program (no. 733176; RabydVax), KU Leuven intramural funding (IOF Hefboom, HB/13/010, and C3, C32/16/039), and the German research foundation (grant no. 391217598, and SFB/TR-237-B14, grant no. 369799452).

Designed experiments, T.V., K.D., H.J.T.; carried out experiments, M.R., J.P., C.C., V.S., L.H., J.T., J.M., and K.G.; analyzed data, M.R., J.M., T.V., and H.J.T.; provided advice on the interpretation of data, K.D., K.K.A., and S.V.G.; wrote the original draft with input from coauthors, M.R. and H.J.T.; wrote the final draft, M.R., T.V., K.D., and H.J.T.; provided and facilitated access to essential samples, N.M., L.-H.L., D.B.K., S.R., H.K., and J.T-S.; supervised the study, K.D. and H.J.T.; and acquired funding, S.R., L.C., J.N., and K.D. All authors approved the final manuscript.

We declare no competing interests.

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
