## [Reviewer comments · Microbiology Spectrum]

Microbiology Spectrum

A high-throughput neutralization assay for yellow fever serodiagnostics

Madina Rasulova, Thomas Vercruyssen, Jasmine Paulissen, Catherina Coun, Vanessa Suin, Leo Heyndrickx, Ji Ma, Katrien Geerts, Jolien Timmermans, Niraj Mishra, Li-Hsin Li, Dieudonné Buh Kum, Lotte Coelmont, Steven Van Gucht, Hadi Karimzadeh, Julia Thorn-Seshold, Simon Rothenfußer, Kevin Ariën, Johan Neyts, Kai Dallmeier, and Hendrik Thibaut

Corresponding Author(s): Hendrik Thibaut, Rega Institute for Medical Research

Review Timeline:

Submission Date:	December 18, 2021
Editorial Decision:	March 2, 2022
Revision Received:	May 19, 2022
Accepted:	May 19, 2022

Editor: Juan Ludert

Reviewer(s): Disclosure of reviewer identity is with reference to reviewer comments included in decision letter(s). The following individuals involved in review of your submission have agreed to reveal their identity: Beate Kampmann (Reviewer #1)

Transaction Report:

DOI: <https://doi.org/10.1128/spectrum.02548-21>

March 2, 2022

Dr. Hendrik Jan Thibaut
Rega Institute for Medical Research
Leuven
Belgium

Re: Spectrum02548-21 (A high-throughput neutralization assay for yellow fever serodiagnostics)

Dear Dr. Hendrik Jan Thibaut:

Thank you for submitting your manuscript to Microbiology Spectrum. Both reviewers found the manuscript of interest and made suggestions to improve some aspects of it. Please take into consideration their comments, especially the doubt about the MOI used when sending the revised version. When submitting the revised version of your paper, please provide (1) point-by-point responses to the issues raised by the reviewers as file type "Response to Reviewers," not in your cover letter, and (2) a PDF file that indicates the changes from the original submission (by highlighting or underlining the changes) as file type "Marked Up Manuscript - For Review Only". Please use this link to submit your revised manuscript - we strongly recommend that you submit your paper within the next 60 days or reach out to me. Detailed instructions on submitting your revised paper are below.

Link Not Available

Sincerely,

Juan E. Ludert

Journals Department
Reviewer comments:

Reviewer #1 (Comments for the Author):

The paper by Rasulova et al describes a new approach to yellow fever serology assays using reporter gene technology. The paper is well written and very accessible and most importantly addresses an unmet need-which is to simplify the very cumbersome cell-culture and microscopy approach required for the currently used plaque reduction assay. This assay is relatively poorly standardised and can only be carried out in a few accredited laboratories.

I have only few specific comments:

In the introduction the authors state that 1 dose of YF vaccine affords life-long protection this is not yet fully established as the

WHO recommendations were only recently changed-please tone this statement down.

Results: they authors state that the fluorescent-tagged transgene used is stable for 6 passages- how would this be addressed if reagents are shared with other laboratories- please address in the discussion and comment on how the stability of the construct is assured.

Please state the numbers of samples used in the he'd-to head comparisons, also in the main manuscript -some of this detail is in the supplementary but should be clear how thorough these comparisons have been, especially with human sera.

define the term assay selectivity- is this not specificity?

Implementation for high throughput- specify how many of the samples were from each species, in particular human sera.

Discussion: first para: "or population-wide monitoring of vaccine efficacy"- you are not talking about vaccine efficacy but immunogenicity here-please correct terminology. There is a lot of repetition from the introduction- cut down as already mentioned in the intro.

Please state the volume of serum required in the new assay compared to the gold standard assay- I don't think, this is spelled out in the manuscript.

Reviewer #2 (Comments for the Author):

In their work, Rasulova and collaborators describe development of neutralization assay for yellow fever serodiagnostic, capable of high-throughput application. Assay is based on use of fluorescently tagged yellow fever virus (strain YF17D). This fluorescence based assay was compared to other neutralization tests; plaque reduction neutralization test (golden standard) and CPE-based 96-well plate neutralization. Assay was also validated by two reference laboratories and test specificity was tested using antibodies to other Flaviviruses (Japanese encephalitis virus, Zika virus, and Dengue type 2 virus). Work is for most part well done and it is well written. It shows capacity to be used for high-throughput screening.

Major comments.

One data which I am missing in text is the multiplicity of infection used for assay. Viral titration described in Material and Method section is inadequate for precise determination of viral infectious particles. In my opinion, it is important to determine titer of stock virus, used for experiments, by classical plaque assay. This would allow to compare amount of infectious viral particles used and number of spots detected. Same method to titrate viruses should be used for other fluorescence modified viruses used in this study.

As alternative, I would also suggest determination of tissue culture infectious dose 50% (TCID₅₀) per ml, then fixed amount of TCID₅₀ could be used. This methodology is very useful for this type of experiments (J Virol Methods. 2018 April ; 254: 1-7. doi:10.1016/j.jviromet.2018.01.003). Authors should comment on their method of quantification, and prove that experiment would be reproducible.

It is not clear why the infection is left to proceed for 3 days. Such long time post infection opens possibility of secondary infection, which could distort results. It is not necessary to have many thousands of spots, with 200 or 400 infected cells in control well would be sufficient to observe clear neutralization effect. This level of observed spots, with high Z value are observed 2 days after infection.

With respect to setting experimental conditions, in Figure S1 is shown how the conditions were chosen. However, immediate lower dilution shows low Z' value, which could cause false results in case there was reduction of virus titer. Higher amount input virus could be beneficial. Please comment.

Furthermore, it is not explained if each detected spot represents one fluorescence cell, as result of infection with one viral particle, or they could be groups of infected cells present.

Minor point.

What does it mean "very comparable" at the bottom of page 5. Results are comparable or not, and comparison could be statistically significant or not. Please remove very.

Staff Comments:

Preparing Revision Guidelines

Please return the manuscript within 60 days; if you cannot complete the modification within this time period, please contact me. If you do not wish to modify the manuscript and prefer to submit it to another journal, please notify me of your decision immediately so that the manuscript may be formally withdrawn from consideration by Microbiology Spectrum.

Point-by-point response to Reviewers:

Reviewer #1 (Comments for the Author):

The paper by Rasulovala et al describes a new approach to yellow fever serology assays using reporter gene technology. The paper is well written and very accessible and most importantly addresses an unmet need-which is to simplify the very cumbersome cell-culture and microscopy approach required for the currently used plaque reduction assay. This assay is relatively poorly standardised and can only be carried out in a few accredited laboratories.

I have only few specific comments:

In the introduction the authors state that 1 dose of YF vaccine affords life-long protection this is not yet fully established as the WHO recommendations were only recently changed-please tone this statement down.

As correctly pointed out by the reviewer, the most recent WHO and CDC guidelines indicate that “for most people, a single dose of yellow fever vaccine provides long-lasting (up to 35 years) protection and a booster dose of the vaccine is not needed. However, travelers going to areas with ongoing outbreaks may consider getting a booster dose of yellow fever vaccine if it has been 10 years or more since they were last vaccinated” (Centers for Disease Control and Prevention, 2021).

We therefore adjusted the introduction replacing the statement of “life-long protection” by “long-lasting protection (up to 35 years) in most vaccinees” (lines 62-63).

Results: they authors state that the fluorescent-tagged transgene used is stable for 6 passages- how would this be addressed if reagents are shared with other laboratories- please address in the discussion and comment on how the stability of the construct is assured.

As described in the Material and Methods section, viral stocks (passage 0) can be produced from pShuttle-YFV/mCherry plasmid transfection (or similar pShuttle constructs for DENV2, ZIKV and JEV) (lines 306-312). These plasmids are stable and can be shipped to other laboratories at ambient temperature. From this passage 0, large YFV/mCherry stocks can be produced that stably express the mCherry for up to at least six passages, as shown by RT-PCR fingerprinting. A step-by-step procedure how to produce a virus stock from this plasmid is provided as Extended Data File.

To further establish whether passaged YFV/mCherry retains similar neutralization capacities, we performed SNT^{FLUO} assays on the WHO International reference sample using YFV/mCherry from six different passages (P0 and P1- P6). These data are now included as Figure S1d and clearly indicate that passaging of YFV/mCherry up to six passages does not affect neutralization results with SNT₅₀ values remaining comparable to those obtained by PRNT.

We also have added a section to the discussion addressing how the stability of our construct is assured when sharing reagents between different laboratories (lines 251-256).

Please state the numbers of samples used in the head-to-head comparisons, also in the main manuscript -some of this detail is in the supplementary but should be clear how thorough these comparisons have been, especially with human sera.

As requested by the reviewer, we have added the information throughout the manuscript on the number of samples used in each experiment (lines 160, 161, 182, 183, 203, 204, 216, 655 and 656).

Define the term assay selectivity- is this not specificity?

As correctly pointed out by the reviewer, we have replaced the term “selectivity” by “specificity” throughout the manuscript.

Implementation for high throughput- specify how many of the samples were from each species, in particular human sera.

We have added the requested information to the manuscript (lines 203, 204 and 216). In total, we tested 867 mouse, 132 hamster, 96 pig, 174 monkey and 727 human serum samples.

Discussion:

First para: "or population-wide monitoring of vaccine efficacy"- you are not talking about vaccine efficacy but immunogenicity here-please correct terminology.

As correctly pointed out by the reviewer, we have replaced the term ‘vaccine efficacy’ by “immunogenicity” (line 224).

There is a lot of repetition from the introduction- cut down as already mentioned in the intro.

As suggested by the reviewer, we have cut down some overlapping sections from the introduction.

Please state the volume of serum required in the new assay compared to the gold standard assay - I don't think, this is spelt out in the manuscript.

According to WHO guidelines, 50µl of serum sample is required for a PRNT assay in a 6-well plate starting at a 1:10 serum dilution (World Health Organization, 2007). Our SNT^{FLUO}, in contrast, only requires 26µl of serum sample to start at the same dilution. This information is now added to the discussion (line 247).

Reviewer #2 (Comments for the Author):

In their work, Rasulova and collaborators describe development of neutralization assay for yellow fever serodiagnostic, capable of high-throughput application. Assay is based on use of fluorescently tagged yellow fever virus (strain YF17D). This fluorescence based assay was compared to other neutralization tests; plaque reduction neutralization test (golden standard) and CPE-based 96-well plate neutralization. Assay was also validated by two reference laboratories and test specificity was tested using antibodies to other Flaviviruses (Japanese encephalitis virus, Zika virus, and Dengue type 2 virus). Work is for most part well done and it is well written. It shows capacity to be used for high-throughput screening.

Major comments.

One data which I am missing in text is the multiplicity of infection used for assay. Viral titration described in Material and Method section is inadequate for precise determination of viral infectious particles. In my opinion, it is important to determine titer of stock virus, used for experiments, by classical plaque assay. This would allow to compare amount of infectious viral particles used and number of spots detected. Same method to titrate viruses should be used for other fluorescence modified viruses used in this study. As alternative, I would also suggest determination of tissue culture infectious dose 50% (TCID₅₀) per ml, then fixed amount of TCID₅₀ could be used. This methodology is very useful for this type of experiments (J Virol Methods. 2018 April ; 254: 1-7. doi:10.1016/j.jviromet.2018.01.003). Authors should comment on their method of quantification, and prove that experiment would be reproducible.

As requested, we have determined the infectious virus titer for YFV/mCherry, DENV2/mCherry and JEV/GFP by classical plaque assay: ~2000 spots/well used in the SNT^{FLUO} assay corresponds to an MOI of 0.02, 0.04 and 4.5 for YFV/mCherry, JEV/GFP and DENV2/mCherry, respectively. Unfortunately,

due to an increased attenuation by the presence of the mCherry insert, ZIKV/mCherry does not induce measurable CPE either by plaque or TCID₅₀ assay. Despite the fact that the SNT^{FLUO} assays with JEV/GFP, DENV2/mCherry and ZIKV/mCherry still need to be further characterized and validated (see also discussion lines 259-261), our data already suggest that they could be implemented in a similar way as the YFV/mCherry SNT^{FLUO} assay. The information on the MOI has now been added to the M&M section (lines 316-318 and 346-348), the figure legends (lines 670 and 675-676), and in the main text (line 126).

In addition, to further demonstrate the assay reproducibility, we performed seven parallel SNT^{FLUO} assays at MOI 0.02 on the WHO International reference sample using YFV/mCherry from different passages (P0, i.e. harvest after transfection, and P1 to P6). These data are now included as Figure S1d and clearly indicate that the method of quantification is accurate and that passaging of the virus does not affect neutralization results with SNT₅₀ values remaining comparable to those obtained by PRNT.

It is not clear why the infection is left to proceed for 3 days. Such long time post infection opens possibility of secondary infection, which could distortion results. It is not necessary to have many thousands of spots, with 200 or 400 infected cells in control well would be sufficient to observe clear neutralization effect. This level of observed spots, with high Z value are observed 2 days after infection.

We agree with the reviewer that secondary infections could occur thereby possibly affecting the neutralization results. However, we compared our assay head-to-head with the conventional PRNT using the WHO reference sample, revealing that, despite possible secondary infections, neutralization curves remain similar (Figure 1e and Figure S1d). Furthermore, we fully agree that it is possible to read out two days post-infection with lower number of infected cells (spots), while maintaining high Z' values. However, this requires more virus input and as a consequence, a more regular production of new virus stocks. In addition, this three-day incubation period is also favorable for practical reasons as it allows us to perform neutralization assays over the weekend.

With respect to setting experimental conditions, in Figure S1 is shown how the conditions were chosen. However, immediate lower dilution shows low Z' value, which could cause false results in case there was reduction of virus titer. Higher amount input virus could be beneficial. Please comment.

Similar as for PRNT, it is key to first determine the optimal virus inoculum that will be used across different experiments, followed by validation with an international reference serum that allows for a direct head-to-head comparison with the gold standard assay. At the conditions chosen, EC₅₀ values obtained by SNT^{FLUO} (irrespective of the passage number) are very much comparable and in the same range as those obtained by standard PRNT (including cross validation by reference laboratories). In addition, and as also described above, the amount of virus used in our SNT^{FLUO} assay is now more standardized by precisely determining the amount of infectious virus particles. This information has also been added to the Extended data protocol.

Furthermore, it is not explained if each detected spot represents one fluorescence cell, as result of infection with one viral particle, or they could be groups of infected cells present.

We have analyzed our SNT^{FLUO} assay both by ImmunoSpot reader and high content imaging to accurately quantify the number of infected cells. In the same surface, we determined that a single spot measured by ImmunoSpot reader corresponds to on average 2.5 infected cells as measured by high content imaging analysis. We have added this information to the main text of the manuscript (lines 135-138).

Minor point.

What does it mean "very comparable" at the bottom of page 5. Results are comparable or not, and comparison could be statistically significant or not. Please remove very.

As correctly pointed out by the reviewer, we removed “very” from the sentence (line 168). “Comparability” s.str. has been demonstrated by extensive correlation analysis (Pearson; Bland-Altman) throughout the manuscript.

References

Centers for Disease Control and Prevention. (2021). Yellow Fever Vaccine Recommendations.

Retrieved from <https://www.cdc.gov/yellowfever/vaccine/vaccine-recommendations.html>

World Health Organization. (2007). Guidelines for plaque reduction neutralization testing of human antibodies to dengue viruses. Retrieved from <https://apps.who.int/iris/handle/10665/69687>

May 19, 2022

Dr. Hendrik Jan Thibaut
Rega Institute for Medical Research
Leuven
Belgium

Re: Spectrum02548-21R1 (A high-throughput neutralization assay for yellow fever serodiagnostics)

Dear Dr. Hendrik Jan Thibaut:

I am pleased to inform you that your manuscript has been accepted, and I am forwarding it to the ASM Journals Department for publication. You will be notified when your proofs are ready to be viewed.

Sincerely,

Juan E. Ludert
Editor, Microbiology Spectrum

Journals Department
Supplemental file 1: Accept